Psychosocial functioning before and after surgical treatment for morbid obesity: reliability and validation of the Norwegian version of obesity-related problem scale

Aasprang Anny 1 2 anny.aasprang@hisf.no
Andersen John Roger 1 3
Våge Villy 4 5
Kolotkin Ronette L. 1 3 6 7
Natvig Gerd Karin 2
1 Faculty of Health Studies, Sogn og Fjordane University Collage , Førde , Norway
2 Department of Global Public Health and Primary Care, University of Bergen , Bergen , Norway
3 Department of Surgery, Førde Central Hospital , Førde , Norway
4 Department of Surgey, Voss Hospital, Helse Bergen Trust , Voss , Norway
5 Centre for Health Research, Førde Hospital Trust , Førde , Norway
6 Department of Community and Family Medicine, Duke University School of Medicine , Durham, NC , USA
7 Quality of Life Consulting, PLLC , Durham, NC , USA
Nock Nora
Electronic publication date: 2015 Sep 29
Publication date: 2015
Volume: 3
Electronic Location ID: e1275
Received 2015 Mar 10; Accepted 2015 Sep 6
Copyright: © 2015 Aasprang et al.
Copyright year: 2015
Copyright holder: Aasprang et al.
License: This is an open access article distributed under the terms of the Creative Commons Attribution License, which permits unrestricted use, distribution, reproduction and adaptation in any medium and for any purpose provided that it is properly attributed. For attribution, the original author(s), title, publication source (PeerJ) and either DOI or URL of the article must be cited.
License URL: https://creativecommons.org/licenses/by/4.0/

Keywords: Obesity, Obesity surgery, Quality of life, OP-scale, Psychosocial functioning, Validity, Reliability, Questionnaires

Funding: Faculty of Health Studies, Sogn og Fjordane University College, Norway The study was financially supported by the Faculty of Health Studies, Sogn og Fjordane University College, Norway. The funders had no role in study design, data collection and analysis, decision to publish, or preparation of the manuscript.

==============================
Background. The aims of this study were to translate the Obesity-Related Problem scale (OP scale) into the Norwegian language and test its reliability, validity and responsiveness in a Norwegian sample.

Method. The questionnaire (OP scale) was translated from the original language (Swedish) into Norwegian. Patients completed the questionnaire prior to and one year after sleeve gastrectomy. Internal consistency was evaluated using Cronbach’s α. Construct validity was tested by correlating the OP-scale with the SF-36 and the Cantril Ladder using the Pearson correlation coefficient. An exploratory and confirmatory factor analysis was used to test the unidimensionality of the OP scale. Responsiveness was tested by assessing changes in the OP scale from baseline to one year post-surgery using the paired sample t-test. Floor and ceiling effect were calculated as percentages.

Results. A total of 181 patients (123 women) accepted for bariatric surgery was included in the study. The mean age was 43.1 ± 12.5 years, and mean body mass index (BMI) before surgery was 45 ± 6.9. The mean value of the OP scale at baseline was 63.30 ± 24.43 (severe impairment) and 21.01 ± 20.98 at one year follow-up (mild impairment). Internal consistency was high at baseline (Cronbach’s α 0.91). The floor effect was small at baseline and high at one year. The ceiling effect was small at baseline and at one year. Exploratory and conformatory factor analysis showed one factor with a high percent of explained variance. Correlations between OP scale at baseline, SF-36, Cantril Ladder and BMI were statistically significant and in the predicted direction to support validity of the Norwegian OP scale. After one year correlations between the change in OP scale and the change in SF-36 scores, Cantril Ladder and BMI were also statistically significant, except for the change in the Role Physical-scale. The OP scale showed greater responsiveness than either the SF-36 or Cantril Ladder.

Conclusion. These results confirm that the Norwegian version of the OP scale is a valid and reliable instrument for measuring psychosocial functioning in patients with clinically severe obesity.

Introduction

Individuals with obesity often report reduced health-related quality of life (HRQL) compared to individuals with normal weight (Fontaine & Barofsky, 2001; Kolotkin, Meter & Williams, 2001; Kushner & Foster, 2000; Larsson, Karlsson & Sullivan, 2002), and improvement in HRQL is one of the commonly stated objectives of surgical treatment of morbid obesity (Munoz et al., 2007). Several studies have shown a great improvement in HRQL after bariatric surgery (Aasprang et al., 2013; Helmio et al., 2011; Karlsson et al., 2007; Kolotkin et al., 2012; Schouten et al., 2011; Zijlstra et al., 2013) and the importance of evaluating HRQL and change in HRQL is underlined.

There are three basic approaches to measuring quality of life: disease-specific measures, generic measures and overall quality of life/life satisfaction. Both generic and disease-specific instruments are utilized to assess the burden of obesity (Fontaine & Barofsky, 2001; Kolotkin, Meter & Williams, 2001; Kushner & Foster, 2000). Generic instruments focus on broad dimensions of health and do not cover all of the domains that are relevant for specific diseases, such as obesity. On the other hand disease-specific instruments are used to capture information that is most pertinent to particular patient groups (Karlsson et al., 2003). Overall quality of life is a subjective assessment of how happy or satisfied a person is with life as a whole (Wilson & Cleary, 1995). In the past decade several obesity-specific HRQL instruments have been introduced (Duval et al., 2006; Kolotkin et al., 2001; Le Pen et al., 1998; Sullivan et al., 1993).

Psychosocial functioning is important in the assessment of HRQL in obesity (Sullivan et al., 1993), and weight related psychosocial distress is not assessed in generic instruments. The Obesity-Related Problems scale (OP scale) was developed in the Swedish Obese Subject Study (SOS) specifically to assess psychosocial problems related to obesity (Karlsson, Sjostrom & Sullivan, 1998; Sullivan et al., 1993). The OP scale is scored so that lower scores represent higher psychosocial functioning. The OP scale has been used in several studies in different countries (Karlsson et al., 2007; Kaukua et al., 2003; Larusdottir et al., 2014; Oh et al., 2013; Sovik et al., 2013), but to our knowledge the OP scale has only been validated in the Swedish, Spanish and Korean languages. Results of these validation studies show that the OP scale has satisfactory reliability and validity (Bilbao et al., 2009; Karlsson et al., 2003; Lee et al., 2013). The OP scale has not been evaluated in surgical patients undergoing sleeve gastrectomy, nor has it been used prospectively, to our knowledge, with the exception of the SOS study.

The aims of this study were to translate the OP scale into the Norwegian language and test its reliability, validity and responsiveness in a Norwegian sample. The specific hypotheses were as follows: (a) the Norwegian OP scale has satisfactory internal consistency; (b) the variability of scores reflect one factor; (c) the OP scale is negatively correlated with both the Cantril Ladder and Short form-36 (SF-36), with the highest correlation coefficient with the social function domain of the SF-36 ; (d) the OP scale is positively correlated with Body Mass Index (BMI); and (e) changes in the OP scale are negatively correlated with changes in both the SF-36 and Cantril Ladder, as well as positively correlated with weight loss as assessed by changes in BMI.

Materials and Methods

Study design and patients

Before the study began we obtained permission from the author of the original Swedish OP scale to develop and validate a Norwegian version of the OP-scale.

The cohort study was performed from 2011 to 2013 in the western region of Norway. A total of 209 patients accepted for bariatric surgery (sleeve gastrectomy) were invited to join the study. The inclusion criteria were age 18–60 years, BMI ≥40.0 or 35.0–39.9 with obesity- related comorbidities, no active psychosis, no drug or alcohol problems, and previous failure to lose weight through other methods. Written informed consent was obtained from participants to complete self-report questionnaires prior to and one year after sleeve gastrectomy. The patients completed the questionnaires at home and brought them to the hospital when they arrived for surgery (baseline) and one-year follow-up. Those who had forgotten the questionnaires were allowed to complete the questionnaires at the hospital. The investigation conforms to the principles outlined in the Declaration of Helsinki. The study was approved by the Regional Committee of Ethics in Medicine, West-Norway (reference number: 2009/2174).

Demographic characteristics and clinical data

Data were collected using a standardized form. Body weight, height, age, gender, educational level, marital status and employment status of the patients were noted. Body weight was measured in light clothing without shoes to the nearest 0.1 kg. Height was measured in a standing position without shoes to the nearest 0.01 m. Body mass index (BMI) was calculated as weight divided by height squared (kg/m2).

Questionnaires administered

The OP scale is an 8-item questionnaire developed for the SOS study to measure the impact of obesity on psychosocial functioning (Karlsson et al., 2003; Sullivan et al., 1993). The OP scales asks respondents to rate on a 4-point scale (“definitely not bothered,” “not so bothered,” “mostly bothered” and “definitely bothered”) if their obesity bothers them in activities such as private gatherings, community activities, and intimate relations. The scale is coded so that lower scores represent higher psychosocial functioning. Scores on individual items are summed to create a raw total score, which can vary between 8 and 32. This score is standardized on a scale from 0 to 100, where 100 indicates the worst possible state and 0 the best possible state. Scores below 20 indicate no or very mild impairment in psychosocial functioning. Scores between 20 and <40 indicate mild impairment, between 40 and <60 moderate impairment, between 60 and <80 severe impairment and 80 or above extreme impairment (Karlsson et al., 2003). We used version 2 of the OP scale.

The Short Form -36 (SF-36) (Norwegian version 1.2) is a well-established generic measure of the health burden of chronic diseases (Ware, 2000). The questionnaire has demonstrated good validity and reliability (Loge & Kaasa, 1998). SF-36 assesses eight dimensions of physical and mental functioning, each ranging from zero (poorest) to 100 (optimal). The subscales physical functioning, physical role function and bodily pain reflect physical functioning, and emotional role function and mental health reflect mental functioning. The subscales general health, vitality and social functioning reflect both physical and mental functioning. The SF-36 can also be divided into two summary scores, Physical Component Summary (PCS) and Mental Component Summary (MCS) (Ware, Kosinski & Dewey, 2000), where a higher score represents better physical or mental health. PCS and MCS scores are standardized so that a difference in 2–4.9 points, respectively, can be interpreted as a small effect size, 5–7.9 points, respectively, a medium effect size and 8+ points a large effect size (Cohen, 1988; Saris-Baglama et al., 2004).

Cantril Ladder is used to assess life satisfaction. The term life satisfaction is often used to describe quality of life, well-being and happiness. Respondents rate their current life satisfaction on a ladder ranging from 10 to 0, where 10 reflects the best life satisfaction and 0 reflects the worst life satisfaction. A score below 6 is considered to be low life satisfaction, and a score of 6 or more is considered to be high life satisfaction (Levin & Currie, 2014).

Translation process

The OP scale was translated from the original language (Swedish) into Norwegian, according to the standards established by the International Quality of Life Assessment Project group (Aaronson et al., 1992; Guillemin, Bombardier & Beaton, 1993). Translation from Swedish to Norwegian was conducted by two individuals whose native language was Norwegian and who have a clear understanding of conceptual meanings in both Norwegian and Swedish languages. The translators were health professionals and were professionally familiar with the concept of morbid obesity. The back translation from Norwegian to Swedish was conducted by two other individuals with an academic background from the social sciences who had Swedish as their native language, as well as a clear understanding of conceptual meanings in both Swedish and Norwegian. The translators worked separately during this phase. A consensus panel of three people compared the original version with the two translated versions and reconciled the forward translations into one common version.

The Norwegian version of the questionnaire was tested on a small sample of patients (n = 8) who had been accepted for bariatric surgery. The aim of the pilot study was to identify and solve any potential problems in the translations, such as confusing words. The patients gave feedback in focus groups composed of four individuals (4 + 4). The questionnaire was found to be easily understood, and no changes to the questionnaire were required.

Statistical analysis

Clinical and sociodemographic data were described as frequency and percentages or means ± standard deviation (SD).

Internal consistency for the OP scale was calculated using Cronbach’s α. Cronbach’s α above 0.7 was considered to be satisfactory (Cohen, 1988). We also calculated Cronbach’s α if one item was deleted. As an addition analysis we correlated the items of the OP score with the total score, and we corrected for overlap. In order to study structure validity, we performed a principal component analysis (PCA) to test whether the items on the OP-scale made up a single factor. The items with factor loading and communality ≥0.40 were considered acceptable (Staquet, Hays & Fayers, 1998). To complement our results, we performed a confirmatory factor analysis (CFA), using the following indexes and cut-offs indicating acceptable fit (Batista-Foguet, Coenders & Alonso, 2004; Devins et al., 2001; Hatcher, 1994; Mulaik, 1989): (a) chi squared divided by the degrees of freedom (χ2/DF) (<2); (b) the root mean squared error of approximation (RMSEA), (<0.08); and (c) the normed fit index (NFI) (>90) and comparative fit index (CFI) (>90). CFA factor loadings ≥0.40 were considered acceptable (Staquet, Hays & Fayers, 1998).

Convergent validity was tested by correlating the OP scale with the SF-36, Cantril Ladder, and BMI, using the Pearson correlation coefficient. A value of <0.1 was considered as trivial, 0.1–0.29 as small, 0.3–0.49 as moderate and values ≥0.5 as large (Cohen, 1988). We tested the association between OP scale and gender by independent T-tests and the association between OP scale and age by Pearson correlation.

A change in OP scale, Cantril ladder and SF-36 from baseline to one year post-surgery was tested using the paired sample t-test. Magnitude of responsiveness was studied by calculating effect sizes (i.e., mean change between assessments, divided by the standard deviation of change) (Cohen, 1988). An effect size <0.2 was considered as trivial, 0.2–<0.5 as small, 0.5–<0.8 as moderate and large ≥0.8 (Cohen, 1988). Comparison of baseline characteristics between responders and non-responders to the follow up was tested by chi-squared (categorical variable) and independent T-test (continuous variables). Floor and ceiling effects were calculated as percentages. Floor or ceiling effects should be below 15% to meet acceptable measurement standards (Wyrwich, Tierney & Wolinsky, 1999).

Given a significance level of 0.05 and a power of 80% we would be able to detect a significant correlation of 0.24 or more between OP scale and other measures when N = 130. Statistical analyses were performed with the statistical program Statistical Package for Social Sciences, for windows, version 22.0 (SPSS, Chicago, Illinois, USA). CFA was conducted using an add-on feature of SPSS Inc. software, AMOS Version 22.0.0.

Results

Figure 1 describes the flow of patients through the study. A total of 209 patients were invited to participate in the study. Baseline analyses were based on 181 patients, and follow-up analyses were based on 130 patients. Among the 181 included patients 123 (68%) were female. Patient characteristics are presented in Table 1. The mean age was 43.1 ± 12.5 years, and mean body mass index before surgery was 45 ± 6.9. The attrition rates (14%) at the 1-year follow-up was higher in women (p = 0.013) and in those who had a lower score on the Cantril Ladder at baseline (p = 0.002).

Figure 1 Flow of patients.

∗ Excluded due to problems with the data-gathering routines. ∗∗ For 26 of the patients we had no post-surgery data because there was less than a year since surgery. Three patients did not meet for follow-up appointment, 21 were excluded due to problems with the data-gathering routines and one had the control with his GP and did not send the questionnaire to the hospital.

Table 1 Characteristics of the patients (n = 181).

Variable	Value	
Age (yr), mean ± SD	43.1 ± 12.5	
Gender, Woman, n (%)	123 (68.0)	
Mean body mass index (kg/m2) baseline, mean ± SD	45.0 ± 6.9	
Current marital status, n (%)		
Married/cohabitants	114 (63.0)	
Live alone	66 (37.0)	
Education, n (%)		
Primary school	42 (23.2)	
High school	96 (53.0)	
University ≤4 y	28 (15.5)	
University >4 y	13 (7.2)	
Notes.

SD standard deviation

Number of patients ranges from 179 to 181.

In Table 2 statistics are presented for OP scale item-total correlations, Cronbach’s α, exploratory factor analysis factor loadings and item communalities at baseline. The patients completed all the items in the OP scale so we had no missing data. Item-total correlations correcting for overlap ranged from 0.53 to 0.80. Cronbach’s α for the OP-scale total score was 0.91. Cronbach’s α if one item deleted ranged from 0.89 to 0.91 for OP scale items. The PCA showed that a single factor explained 62.2% of the variance in the OP-scale. Factor loadings ranged from 0.62 to 0.86 and item communalities from 0.39 to 0.74. The CFA showed that the modification indices (M.I) indicated a high covariance between item 1 and 2 (M.I = 21.37, Par change = 0.135). This could be related to the items having quite similar wording about attending a party and that the items are placed after each other in the questionnaire. Thus, we let the error terms of item 1 and 2 covariate in the model resulting in an improved model fit. The results of the CFA were as follows: χ2/DF = 1.89, RMSEA = 0.070, NFI = 0.96 and CFI = 0.98. Factor loadings ranged from 0.55 to 0.86 (Ps < 0.001).

Table 2 Reliability analyses and exploratory and confirmatory factor analyses (n = 181).

Standarized Cronbach’s α of OP scale baseline 0.91. The PCA showed that a single factor explained 62.2% of the variance in the OP-scale.

Item	Item description	Item-total correlationa	α if one item deleted	Exploratory factor analysis loading	Exploratory factor analysis communality	Confirmatory factor analysis loading	
1	Private gatherings in my own home	0.69	0.90	0.77	0.60	0.70	
2	Private gatherings in a friend’s home	0.80	0.89	0.86	0.74	0.82	
3	Going to a restaurant	0.80	0.89	0.86	0.74	0.86	
4	Going to community activities	0.72	0.90	0.84	0.70	0.83	
5	Holidays away from home	0.78	0.89	0.84	0.71	0.83	
6	Trying and buying clothes	0.66	0.91	0.74	0.54	0.69	
7	Bathing in public places	0.53	0.91	0.62	0.39	0.55	
8	Intimate relations with partner	0.67	0.90	0.745	0.56	0.69	
Notes.

a The item total correlation with its own OP scale correcting for overlap.

At baseline the correlation coefficients between the OP scale and all the self-reported measures and BMI were statistically significant and in the predicted direction (Table 3). We found no association between OP scale and age and gender (data not shown). After one year, the correlation coefficients between the change in OP scale and the change in self-reported measures and change in BMI were also statistically significant, except for the change in the RP-scale (Table 3).

Table 3 Correlations between OP scale, SF-36 and Cantril Ladder at baseline and between changes in these scores after one year.

	OP Baseline (n ranged from 175 to 181)	Δ OP (n ranged from 124 to 130)	
Life satisfaction Baseline	−.561 (p < 0.001)		
BMI Baseline	.186 (p = 0.012)		
SF-36			
PCS Baseline	−.410 (p < 0.001)		
MCS Baseline	−.624 (p < 0.001)		
Physical function	−.321 (p < 0.001)		
Physical role function	−.268 (p < 0.001)		
Bodily pain	−.299 (p < 0.001)		
General health	−.367 (p < 0.001)		
Vitality	−.460 (p < 0.001)		
Social function	−.582 (p < 0.001)		
Emotional role function	−.373 (p < 0.001)		
Mental health	−.570 (p < 0.001)		
Δ Life satisfaction		−.394 (p < 0.001)	
Δ BMI		−.280 (p < 0.001)	
Δ SF-36			
Δ PCS		−.248 (p = 0.006)	
Δ MCS		−.339 (p < 0.001)	
Δ Physical function		−.266 (p = 0.002)	
Δ Physical role function		−.091 (p = 0.306)	
Δ Bodily pain		−.193 (p = 0.028)	
Δ General health		−.229 (p = 0.009)	
Δ Vitality		−.255 (p = 0.004)	
Δ Social function		−.328 (p < 0.001)	
Δ Emotional role function		−.221 (p = 0.013)	
Δ Mental health		−.250 (p = 0.004)	
Notes.

Δ change

SD standard deviation

PCS physical component summary

MCS mental component summary

Since a higher score on the OP scale indicates poorer psychosocial functioning, the correlation between change in OP scale and change in SF-36 and Cantril Ladder is negative.

Data are given as Pearson’s r correlation coefficients. P values < 0.05 were considered statistically significant.

The mean value of the OP-scale was 63.30 ± 24.43 (severe impairment) at baseline and 21.01 ± 20.98 (mild impairment) at 1 year (Table 4). Mean scores for the SF-36 and Cantril Ladder are also presented in Table 4, as well as results of the paired samples t-test to evaluate the responsiveness of changes in the OP scale following surgery. The OP scale had a higher responsiveness (ES 1.7) than the SF-36 (PCS 1.5, MCS 1.0) and Cantril Ladder (ES 1.4). The percentage of patients scoring at the lowest possible level (floor effect) was 1.1% at baseline and 20% at 1 year. The percentage score at the highest possible level (ceiling effect) was 3.9% at baseline and 0% at 1 year.

Table 4 Mean score on OP scale, SF-36 and Cantril Ladder at baseline and at one year post-surgery.

Scores	Baseline Mean (SD) (n = 130)	1 year post-surgery Mean (SD) (n = 130)	p-value	ES	
OP scale	63.30 (24.43)	21.08 (20.98)	<0.001	1.7	
Cantril Ladder	5.01 (1.81)	7.49 (1.51)	<0.001	1.4	
SF-36					
PCS	37.41 (9.56)	51.90 (8.93)	<0.001	1.5	
MCS	42.82 (10.40)	53.35 (9.44)	<0.001	1.0	
Physical function	58.19 (21.99)	88.51 (16.68)	<0.001	1.4	
Physical role function	41.03 (38.03)	80.96 (31.56)	<0.001	1.1	
Bodily pain	49.44 (24.63)	69.92 (26.16)	<0.001	0.8	
General health	46.99 (20.01)	78.36 (19.51)	<0.001	1.6	
Vitality	35.69 (18.32)	61.23 (22.02)	<0.001	1.4	
Social function	64.60 (28.38)	88.65 (19.02)	<0.001	0.8	
Emotional role function	64.06 (39.60)	89.58 (27.35)	<0.001	0.6	
Mental health	69.84 (14.72)	82.12 (15.43)	<0.001	0.7	
Notes.

SD standard deviation

ES effect size

PCS physical component summary

MCS mental component summary

Discussion

Our aim was to translate the OP scale into the Norwegian language and to tests its psychometric properties and responsiveness in a group of severely obese Norwegians prior to and one year after bariatric surgery. The study shows that the Norwegian version of the OP scale is a reliable and valid instrument. The SOS study showed that the OP scale’s psychometric properties were strongly supported, and our results reproduced the same good performance in terms of validity and reliability (Karlsson et al., 2003).

It has also been shown that the OP scale is valid and reliable for use in Spain and Korea (Bilbao et al., 2009; Lee et al., 2013). Our study population is similar to the study population in other validation studies of the OP scale (i.e., bariatric surgery patients) (Bilbao et al., 2009; Karlsson et al., 2003). The mean OP score in our sample prior to surgery was similar to the Korean validation study (Lee et al., 2013) but a little higher than in the original Swedish study and in the Spanish validation study (Bilbao et al., 2009; Karlsson et al., 2003).

In our study, the internal consistency was above the recommended value >0.70 (Nunnally & Bernstein, 1994) which confirms the hypothesis that the Norwegian version of the OP scale has satisfactory internal consistency. This is a similar result as in other validation studies of the OP scale (Bilbao et al., 2009; Karlsson et al., 2003; Lee et al., 2013).

The factor analysis results confirm the unidimensionality of the OP scale. The majority of the total variance is explained by one factor with a high percentage of the variance explained by this factor, similar to that found by the authors of the original questionnaire (Karlsson et al., 2003) and in the Spanish version (Bilbao et al., 2009). The convergent validity of the OP scale was assessed by examining the relationship between the OP scale and SF-36 and Cantril Ladder. High levels of convergent validity were found. The OP scale had significant negative correlation with all eight domains and the two summary measures of SF-36 and also for Cantril Ladder at baseline. The OP scale had a lower correlation coefficient with role physical and higher correlation coefficient with the social function domain. This is not surprising given that OP scale measures one aspect of psychosocial functioning. These last findings have been reported by Bilbao et al. (2009). BMI showed a significant positive correlation with the OP scale, as was found in other studies (Bilbao et al., 2009; Karlsson et al., 2003).

We found significant negative correlations between changes in OP scale and changes in the SF-36 (all the domains and the two summary scores), except for physical role function. This is likely explained by the response categories for physical role which have a low degree of precision (yes versus no). As far as we know, there have not been any previously published studies describing correlations of change scores for the OP scale. Change in life satisfaction was also significant negative correlated with the OP scale, which means that there is a strong relationship between life satisfaction and psychosocial functioning.

Responsiveness of change in OP scale and other self-report measures was analysed by comparing changes in OP scale, SF-36 and Cantril Ladder at baseline and one year after surgery. The OP scale was more responsive to change compared than the SF-36 and Cantril Ladder. These findings are similar to the OP scale in Sweden, where the questionnaire was developed (Karlsson et al., 2003). Other validation studies have not tested the responsiveness before and after surgery (Bilbao et al., 2009; Lee et al., 2013).

Baseline floor and ceiling effects of the OP scale were small, similar to the original version. It is desired that ceiling and floor effect should be minimal (Karlsson et al., 2003). We found floor effects at 20% at one-year follow-up (20% of participants scored at the best possible state). This might suggests that the OP scale might lack the ability to capture changes occurring over time for given individuals. However it is perhaps more likely that having the best possible score actually means that the patients have no problems.

The OP scale measures the impact of obesity on psychosocial functioning. A limitation of the present study is that the OP scale was only validated in a group of patients that had been accepted for bariatric surgery (BMI ≥35), and it is therefore unknown if the Norwegian OP scale is equally valid, reliable, and responsive in other groups of individuals with obesity, for example patients with a BMI between 30 and 35. It is unclear how the small attrition rate (14%) influenced the results. A strength of the study is, however, that we studied responsiveness from baseline to one year after surgery, and it also strengthens the study that we compared the OP scale with well validated HRQL instruments. Finally, as far as we know, this is the first study that has used correlation of change scores in the validation of OP scale.

In conclusion, this Norwegian version of the OP scale is a valid and reliable instrument for measuring psychosocial functioning in a sample with clinically severe obesity in Norway.

We thank L Schjelderup from the Department of Surgery, Førde Central Hospital, for assisting with the data collection.

Additional Information and Declarations

Competing Interests

Author Contributions

Human Ethics

Data Availability

Ronette L Kolotkin is an employee of Quality of Life Consulting.

Anny Aasprang conceived and designed the experiments, performed the experiments, analyzed the data, contributed reagents/materials/analysis tools, wrote the paper, prepared figures and/or tables, reviewed drafts of the paper.

John Roger Andersen conceived and designed the experiments, analyzed the data, contributed reagents/materials/analysis tools, wrote the paper, prepared figures and/or tables, reviewed drafts of the paper.

Villy Våge and Ronette L. Kolotkin reviewed drafts of the paper.

Gerd Karin Natvig conceived and designed the experiments, reviewed drafts of the paper.

The following information was supplied relating to ethical approvals (i.e., approving body and any reference numbers):

The study was approved by the Regional Committee of Ethics in Medicine, West-Norway (Reference number 2009/2174).

The following information was supplied regarding data availability:

http://dx.doi.org/10.5281/zenodo.15961.

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
