# Peer review of "Psychosocial functioning before and after surgical treatment for morbid obesity: reliability and validation of the Norwegian version of obesity-related problem scale"

_PeerJ, doi:10.7717/peerj.1275_

## Round 0.1 · original submission · Major Revisions

Please address issues raised by reviewers and provide a point by point response to each issue noting how and where in the revised manuscript the issue was addressed.

·

Basic reporting

No comment.

Experimental design

No comment

Validity of the findings

In the result section it is stated that "Cronbach's a ranged from 0.89 to 0.91 for Op scale items at baseline..." This is a confusing statement to the reader. It sounds like you suggest that every item can be given a Cronbach's a value. To my knowledge, Cronbach's a describes a relation between items in a scale and is not something inherent in a single item. The figures you present corresponds to alpha if one item was deleted (Table 3 - column 4 and 7), i.e. suggesting that reliability in OP would not improve by deleting an item. Is that what you mean? Please explain more/ present in a clearer way to help the reader understand.

You repeat in some places that a high percentage of the variance was explained by one factor. Please include the exact percent in the result section.

Additional comments

This study translates the OP scale to Norwegian and tests the scales reliability, validity and responsiveness to change in a bariatric surgery sample. The study is generally well conducted and the article is well written.

My major comments are presented under validity of the findings and these should be addressed before publication. I have a few other comments/reflections that I have marked as notes in the enclosed document.

·

Basic reporting

GENERAL

It is an interesting paper about the psychometric properties of the Norwegian version of the Obesity-related problem scale. The authors conducted a translation process into Norwegian, and studied the reliability, validity and responsiveness of the questionnaire by means of traditional statistical procedures. However, there are some areas which the authors have not adequately addressed, and are confusing. They are as follows:

REVISIONS

Introduction

1. Page 2, 2nd paragraph: It is stated that there are three basic approaches to measure quality of life: disease-specific measures, generic measures, and overall quality of life/life satisfaction. Then, you explain disease-specific and generic measures, but there is no mention to overall quality of life. Could you explain what do you mean with this third approach? Which is the difference with generic measures?

2. Page 3, 2nd paragraph: change “SF-36” by “Short form-36 (SF-36)” because it is the first time it appears in the text.

Material and Methods: questionnaires administered

3. Page 5, 1st paragraph: Indicate the reference of which these cut-offs are obtained, as in the Karlsson et al the cut-offs are different.

4. Page 5, 2nd paragraph: It would be necessary to include the reference for the interpretation of the eight domains of the SF-36, since Ware et al. established another interpretation, dividing the eight dimensions into two groups, physical and mental.

5. In general, how are the questionnaires administered at baseline? And how at one year post-surgery? Who administered the questionnaires? What procedures do you used to increase response rate?

Material and Methods: statistical analysis

6. Page 7, 1st paragraph: Add “frequency” to percentages.

7. Page 7, 2nd paragraph: Reliability is measured by internal consistency, but for each item? It is wrong. The internal consistency is measured for a domain. Then, it is possible to calculate Cronbach Alpha of the scale after deleting each item, to see if the reliability improves after deleting an item. But, the internal consistency must be measured of a scale, and not for an item. Further, the reference for Alpha Cronbach must be included. And, which value of Cronbach alpha is considered adequate for reliability? This should be included in the text, if not it is difficult to interpret the results.

8. Page 7, 2nd paragraph: the statistical analysis in this second paragraph is a little messy. After internal consistency, and item total correlation, floor and ceiling effect is mentioned; however this last analysis is more related to responsiveness. After, it is explained construct validity by means of convergent validity and structural validity, but in the middle, the sample size needed is mentioned. Please, sort the statistical analysis section.

9. Page 7, 2nd paragraph: regarding item total correlation, is it correcting for overlap? Explain this point. Further, which value will you consider acceptable? It is fundamental to interpret the results.

10. Page 7, 2nd paragraph: Why is sample size estimated based on the convergent validity and not in other kind of analysis? Why do you consider a correlation of 0.24 adequate for convergent validity? It is a very low value, much lower than the minimum requirements established by psychometric.

11. Page 7, 2nd paragraph: which values do you establish as adequate for floor and ceiling effect? From which threshold do you consider there is a problem? Without this information it is difficult to interpret the results.

12. Page 7, 2nd paragraph: Construct validity, is studied by means of convergent validity with correlations with other instruments such as SF-36, Cantril Ladder and BMI. However there is no hypotheses about between which domains you hope to find a high correlation (convergent validity), or if possible, between which scales you hope to find lower correlation (divergent validity). Without this information the analysis and the validity of the results are incorrect. In addition, you should state the thresholds above which you would consider a result satisfactory, and the reference for this threshold should be included.

13. Page 7, 2nd paragraph: Regarding exploratory factor analysis, how is it possible to use varimax rotation to study unidimensionality of the scale? Further, there is missing information on the interpretation of the results of exploratory factor analysis. Which values are you taking into account for the interpretation? Factor loading? Communalities? Both of them? The variance explained? Which thresholds are you considering for these values? Explain please how you will interpret the results of the exploratory factor analysis.

14. Page 7, 2nd paragraph: As this is a validation of a translated instrument, it would be more appropriate to study the construct validity of the instrument by means of a confirmatory factor analysis, in addition to the exploratory analysis. When an instrument is translated, it is more appropriate to confirm the structure established by the original authors by means of confirmatory factor analysis.

15. Page 7, 2nd paragraph: regarding responsiveness, how do you interpret the results of the ESs? Insert thresholds for the interpretation, and the corresponding reference. If not, it is difficult to interpret the results. Further, did you include any transitional question in the questionnaire at one year to know if patients have improved or not their psychosocial functioning. To study responsiveness of an instrument it is necessary to know if patient has improved or deteriorated its state. Sometimes it is necessary to study the responsiveness according to transitional questions. And regarding responsiveness, comparison of baseline characteristics between responders and non-responders to the follow up should be included.

16. Page 7: known-groups validity could be included.

17. Page 7: which statistical software did you used for the analyses?

Results

18. Page 8: In general, the results are messy. They should be explained in the same order as the statistical analysis section. That is, reliability, validity, and responsiveness.

19. Page 8, 2nd paragraph: it is stated “as expected,…” but in the statistical section in methods there is no comment regarding this sentence. It should be included.

20. Page 8, 3rd paragraph: as stated before, the results are messy. In this third paragraph reliability, and construct validity are mixed, with baseline data and with follow-up data.

21. Page 8, 3rd paragraph: in general, how many patients responded the option “not applicable” to item 8? How do you handle this data? How do you handle missing data in the OP scale? Which is the final sample size for the factor analysis?

22. Page 8, 3rd paragraph: How do you explain the difference of the item 8 factor loading at baseline and at one year? Further, add the information of communalities and the % of explained variance.

23. Page 8, 3rd paragraph: in general, which is the objective of performing all the analyses with baseline data and with follow-up data? It is not explained in the methods section.

24. Page 8, 3rd paragraph: correct the sentence “exploratory factor analysis factor loadings”. Correct “0.52 to 0.80 at”.

25. Page 8, 3rd paragraph: The sentence “Cronbach’s alpha ranged from 0.89 to 0.91 for OP scale items at baseline and from 0.85 to 0.88 at one-year follow-up” is completely wrong. The cronbach alpha is not estimated for an item. Please, clarify this point.

26. Page 8, 4th paragraph: Floor and ceiling effects should be explained with responsiveness, since the effect of these two effects are very important when studying responsiveness. Further, it is stated “low floor effect” and “moderate floor effect”, but in statistical section these thresholds are not explained.

27. Page 8, 5th paragraph: it is stated that correlation between OP scale and age and gender was studied but it was not explained in statistical section. Further it is not correct to study the relationship between the OP scale and gender by means of correlations. Regarding the convergent validity, a statistically significant correlation is not valid for psychometric perspective. A correlation coefficient can be significantly different from 0 but it can be very low and insignificant. The thresholds for interpretation should have been included.

Discussion

28. Page 9, 4th paragraph: The sentence “The item internal consistency” is not correct. The internal consistency is studied for a scale, not for an item. Please clarify this point. Be careful with the cut-off of an alpha cronbach greater than 0.90 as excellent, because it could be a problem as high values, indicating redundancy in the items of the scale.

29. Page 10, 1st paragraph: Regarding factor analysis, there is a sentence about the variance explained by the factor but this information is not shown in results section. All the conclusions regarding exploratory factor analysis are based on results not shown in results section. Apart from that, the unidimensionality should be confirmed by confirmatory factor analysis, or rasch analysis.

30. Page 10, 2nd paragraph: Regarding floor and ceiling effect, it is true that it has a great effect on responsiveness but it depends on the direction of the scale and the moment. I am not agreed with the sentence of “the moderate floor effects at one year follow-up suggest that the OP scale might lack the ability to capture changes occurring over time”. Why? Please clarify this point.

31. Page 10: the discussion, as the results section is also messy. You start discussing reliability, then construct validity, after responsiveness, and later you discuss again another part of validity. Please order the discussion section.

32. Page 10, 3rd paragraph: you discuss about concurrent validity but this term should be included in the statistical section. And, is it really concurrent validity or convergent validity?

33. Page 11, 2nd paragraph: the sentence “responsiveness before and after surgery” is not correct.

Tables and Figures

34. Table 1. Change a “,” by “.”. Why are the numbers rounded to 0 decimals? Add sd as footnote.

35. Table 2: Correct parenthesis for bodily pain. Insert SF-36 in the table. Insert the abbreviations of SD, MCS, PCS as footnote. Insert sample size in the table. Which is the sample size in this case?

36. Table 3. Insert sample size for the analysis in each case. Is item total correlation correcting for overlap? Add data for communalities and the % of explained variance.

37. Table 4. Insert the sample size for each case. Insert SF-36 in the table. Insert the abbreviations as footnotes. Explain somewhere that data are presented as correlation coefficient (and p-value). Correct some p-values please. (p<0.012??). Change “,” by “.”.

Experimental design

No Comments

Validity of the findings

No Comments

Additional comments

No Comments

---

## Round 0.2 · Minor Revisions

Please fully address the previous and new concerns raised by reviewers and provide a point by point response indicating how and where in the revised manuscript each respective issue was addressed.

·

Basic reporting

No comments.

Experimental design

No comments.

Validity of the findings

Please comment on how you think the attrition could have influenced the results. Is there a risk that post-op improvements are inflated due to the attrition as there is a significant association between life satisfaction and OP?

Additional comments

The article has been further improved.

It has been hard to follow the changes in the document as not all changes are visible. It would have been preferable to see what you have deleted and which paragraphs that have been moved around.

My major comments have been addressed.

Not all my remarks added as notes in the first review have been attended. Please do so. As an example the sentence starting with “The mean scores…” line 241-243 has not been changed. You write that you present results from t-test to evaluate responsiveness of change, but you have actually used ES to evaluate responsiveness of change. With t-test you test whether there is a significant difference between two assessment points and then you have used ES to estimate the magnitude of responsiveness. You have also evaluated responsiveness of change in all used questionnaires and not just OP. Consider how you use the word responsiveness throughout the article and be specific about when you report significant change and when you report responsiveness.


You sometimes refer to morbid obesity and sometimes to severe obesity, maybe you could be more consistent in which term you use.

OP is measuring a specific aspect of psychosocial functioning (obesity related/weight related psychosocial functioning). In some places this could be more clearly stated.

Line numbers from the tracked change document,
Abstract: Ceiling effect high at 1 y?
Line 32-34: The new sentence about overall QoL could preferably be put later in the paragraph where the two other measures are defined.
Line 40: Remove a .
Line 45: Remove Psychosocial.
Line 98-100: Numbers are overlapping. A scale score of 20, 40, 60 or 80 belongs to two categories.
Line 161: A value of 0.5 belongs to no category. Change > to ≥?
Line 167-169 Correct brackets.
Line 168: A n is missing.
Line 381: Change to capital J.
Table 1: University, categories are overlapping regarding 4 ys.

·

Basic reporting

No comments

Experimental design

No comments

Validity of the findings

No comments

Additional comments

GENERAL

The paper has improved greatly and the authors have adequately responded to all questions. I appreciate the effort made by the authors. After these changes, I have only some minor comment.

Minor revisions

Material and Methods: questionnaires administered

1. Line 135. It would be more appropriate “…ranging from zero (poorest) to 100 (optimal).”

Material and Methods: statistical analysis

2. Line 175. It would be desirable to include the reference of Cronbach.

3. Line 178 – Line 186. It is stated that exploratory factor analysis and confirmatory factor analysis is performed. And in the next paragraph (line 188-line 192) the authors address the analysis performed to study construct validity. However, the exploratory and confirmatory factor analyses are focused to study the structure validity of an instrument, and the structure validity is one of the most important type of validity within construct validity. I think that this structure validity should be part of the construct validity.

4. Line 204. I still do not understand the estimation of the sample size. ¿The correlation between…?

Discussion

5. Line 269. As in statistical section, the discussion about factor analyses should be included as part of construct validity.

Tables and Figures

6. Table 2: The title of the table should be extended since Table 2 includes results of the factor analyses, reliability,..

7. Table 4. Regarding the results of responsiveness, why are the baseline mean values estimated from the total sample, and not from the responders to the follow-up? For this analysis only patients with baseline and follow-up should be included.

---

## Round 0.3 · Minor Revisions

Although the majority of issues have been addressed adequately, there is still concern regarding one issue invoking the overlap of categories (Line 129-130) - please address this issue.

·

Basic reporting

No comments

Experimental design

No comments

Validity of the findings

Please re-write the sentence about attrition. It sounds like you are guessing about the effects of attrition rather than discussing possible effects of attrition.

Additional comments

The manuscript is further improved and I think that it could be accepted for publication after correcting the remaining comments.

Line 129-130 Categories in OP are still overlapping. 40 and 60 belongs to two categories.

---

## Round 0.4 · accepted · Accept

The comments have all been addressed